# COVID-19 Lockdowns Promoted Recovery of the Yangtze River's Aquatic Ecosystem

Dongdong Fang [1,2], Haile Yang [2], Hui Zhang [2], Hao Du [2], Junlin Yang [2] and Qiwei Wei [1,2,*]

1   Wuxi Fisheries College, Nanjing Agricultural University, Wuxi 214081, China
2   Key Laboratory of Freshwater Biodiversity Conservation, Ministry of Agriculture and Rural Affairs, Yangtze River Fisheries Research Institute, Chinese Academy of Fishery Sciences, Wuhan 430223, China
*   Correspondence: weiqw@yfi.ac.cn; Tel.: +86-027-8178-0118

**Abstract:** The impacts of COVID-19 lockdowns on human life, air quality, and river water quality around the world have received significant attention. In comparison, assessments of the implications for freshwater ecosystems are relatively rare. This study explored the impact of COVID-19 lockdowns on aquatic ecosystems in the Yangtze River by comparing river water quality, phytoplankton, zooplankton, and fish data collected at the site in the middle reach of the Yangtze River in 2018 and 2020. The results show that during COVID-19 lockdowns, the reduction in industrial and domestic effluent discharge led to a reduction in organic pollution and industrial plant nutrient pollution in rivers. Among them, $PO_4^{3-}-P$, $COD_{Mn}$, and TP were significantly decreased ($p < 0.05$). During lockdowns, nutrient supplies such as TN and TP were reduced, which led to inhibition of algae growth and decreased phytoplankton abundance. Phytoplankton affects the abundance of zooplankton through a bottom-up effect, and a decrease in phytoplankton density leads to a decrease in zooplankton density. The decrease in plankton density led to lower primary productivity in rivers, reduced fish feed supplies, intensified competition among fish populations, with increases in population dominated by high trophic level carnivorous fish. In addition, the decrease in fishing intensity has contributed to an increase in the number of rivers–sea migratory fish; the fish community was earlier mainly dominated by small-sized species with a short life cycle, and the number of supplementary populations has now increased. As a consequence, the fish community structure shows a tendency toward high complexity and high fish diversity. Overall, these observations demonstrate that the rapid revival of the retrogressive Yangtze River ecosystem is possible through limitation of anthropic interferences.

**Keywords:** COVID-19 lockdowns; freshwater ecosystems; human activities

## 1. Introduction

Freshwater ecosystems are among the most threatened in the world [1], and the loss of biodiversity in freshwater systems exceeds that of terrestrial and marine environments [2]. Freshwater biodiversity is widely acknowledged as being in crisis [3]; according to the International Union for the Conservation of Nature (IUCN) Red List of Threatened Species, one in three freshwater species is under threat. The rate of decline of vertebrate populations is much higher in freshwater (81%) than in terrestrial (38%) and marine (36%) environments [4]. After amphibians, freshwater fish species are the most threatened group of vertebrates. Freshwaters—lakes, rivers, wetlands, including floodplains—have always played a major role in the history of humankind and the goods and services they provide are of key importance to our survival and well-being. For example, food supply, water purification, flood regulation, carbon sequestration, transportation, etc. [5] However, freshwater biodiversity is experiencing unprecedented and growing pressure from human activities [6]. At the same time, more than 60% of freshwater habitats are moderately or highly threatened due to human overexploitation [7]. Threats to freshwater ecosystems by human activities are many and varied, spanning long-standing [8] as well as emerging [9]

threats. Therefore, assessing the impact of human activities on freshwater ecosystems is of great significance for the conservation and restoration of freshwater ecosystems.

The coronavirus pandemic, also known as the COVID-19 pandemic, took over the world almost instantaneously and has impacted ~220 countries or territories around the world. It is one of the deadliest pandemics in history [10]. According to the WHO COVID-19 Dashboard (https://covid19.who.int/) (accessed on 3 November 2022), as of 2 November 2022, 628 million people have been infected with the coronavirus leading to over 6.57 million deaths worldwide. COVID-19 is principally spread via respiratory tract with high infectivity. Droplet transmission is the predominant route of contagion from person to person [11]. However, there are other routes besides respiratory transmission. Studies reported SARS-CoV-2 shedding in human stool [12], and therefore in wastewater. Such as suspended particles in wastewater not only can serve as reservoirs for SARS-CoV-2, but also maintain its activity by defending it from the oxidizing agents prevailing in wastewater [13]. Currently, the fate of SARS-CoV-2 in WWTP and its ultimate removal at different stages of treatment remains unexplored and requires urgent attention, especially where treated wastewater is utilized as reclaimed water [14]. To slow down diffusion of the virus, more than 170 countries imposed different modalities of nationwide lockdowns [15]. During lockdown periods, industries and businesses were shut down, and large portions of society were isolated, reducing the regional and global movements of people [16]. The implementation of worldwide lockdowns offers an unprecedented opportunity to study the impact of anthropogenic activities on the natural ecosystem [17]. Some studies, which examined restricted human activities during lockdowns, also concluded that lockdowns positively impacted riverine water and other aquatic ecosystems. For example, studies have reported that China's COVID-19 lockdowns significantly reduced industrial pollution and improved the quality of the water environment [18]. In India, a reduction in pollution and shipping traffic improved the habitat quality in the Ganges River, facilitating spawning migrations of the anadromous hilsa (*Tenualosa ilisha*) (Hamilton, 1822) [19]. In that study, it was found that artificial noise associated with shipping was significantly reduced during the lockdown, thereby increasing the communication ranges of fish and dolphins by 65% [20]. Many studies across the world have suggested that COVID-19 lockdowns improved air quality by reducing levels of nitrogen dioxide ($NO_2$), carbon dioxide ($CO_2$), and particulate matter (PM2.5 or PM10) [21–23]. Freshwater ecosystems are closely linked with and therefore more vulnerable to human activity [24]. However, there has been less research on how to quantify the impact of COVID-19 lockdowns on freshwater ecosystems.

The Yangtze River is the third longest and third most water-rich river system in addition to being one of the most human-impacted of the large rivers in the world [25] due to overfishing, water pollution, species invasion, noise pollution, navigation, urbanization, lake reclamation, etc. [26]. As the first country impacted by COVID-19, China implemented local lockdowns that shut down industries and required people to stay at home between January and May 2020. COVID-19 lockdowns reduced the high intensity of human activity, and we predicted that less shipping traffic, lower fishing intensity, and fewer pollution emissions would have contributed to restoration of the Yangtze River aquatic ecosystem. In this study, we sought to quantify the impact of human activities on the aquatic ecosystem of the Yangtze River based on river water quality, phytoplankton, zooplankton, and fish data collected at the site in the middle reaches of the Yangtze River in 2018 and 2020. The findings provide a basis for the sustainable development of fish resources and a scientific foundation for the protection and management of Yangtze River aquatic ecosystems.

## 2. Materials and Methods

### 2.1. Study Area

The middle reaches of the Yangtze River, from Yichang in Hubei Province to the Hukou of Jiangxi Province, are 955 km in length and cover an area of 68 km$^2$ [27]. The middle reaches of the Yangtze River are one of the regions of high population density in China. The basin is home to more than 0.13 billion people and generates approximately

46.9% of China's gross domestic product (GDP). Moreover, the Yangtze River is the busiest river in the world in terms of inland vessel navigation, with numerous diverse watercrafts ranging from bamboo rafts to large cruise ships. In recent decades, the aquatic biodiversity of the middle reaches of the Yangtze River has been seriously threatened due to the rapid development and modernization of China's economy [28] through damming, reclamation of lakes for farmland, waterway channel construction, and vessel navigation [29]. In this study, the Shishou (SS) and Jiayu (JY) reaches, which are middle reaches of the Yangtze River, were selected as the research areas (Figure 1). On the one hand, the SS and JY river reaches are important survey points for fishery resources and the environment in the middle reaches of the Yangtze River. The Yangtze River Fisheries Research Institute, Chinese Academy of Fishery Sciences, has been conducting year-round surveys at these locations since 2017 [30]. On the other hand, SS and JY, as important cities in the "Yangtze River Economic Belt", have made significant contributions to the rapid development of China's economy [31]. They are also one of the most heavily polluted urban-based river systems [32].

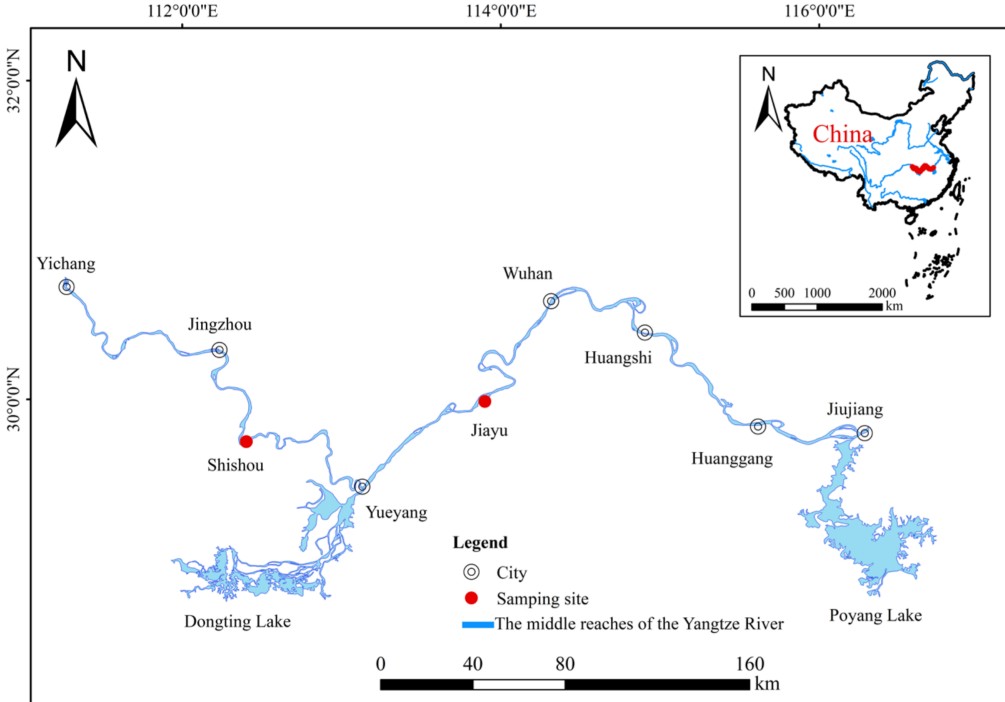

**Figure 1.** Sampling sites in the middle reaches of the Yangtze River.

### 2.2. Data Collection

Samples were collected in July 2018 and July 2020. Water temperature (WT), dissolved oxygen (DO), pH, and oxidation–reduction potential (ORP) were determined in situ at each sampling site using a portable multiparameter water quality instrument (YSI Pro-Plus, YSI Inc., Yellow Springs, OH, USA). Three parallel water samples (1 L each) were collected at various points and stored in an incubator at a low temperature (4 °C). The samples were brought back to the laboratory for water quality index measurement within 48 h [33] to assess total nitrogen (TN), total phosphorus (TP), phosphate ($PO_4^{3-}$-P), nitrate ($NO_3$-N), potassium permanganate index ($COD_{Mn}$), and ammonia nitrogen ($NH_4^+$-N) based on *Standard Methods for the Examination of Water and Wastewater* [34].

Qualitative samples of phytoplankton were collected using a 25# plankton net; quantitative samples were collected using 1 L water collectors, and Lugol's reagent was added to fix the sample after collection, which was then concentrated to 50 mL for 48 h in the dark for identification and counting. Qualitative samples of zooplankton were collected

using 13# plankton nets, fixed with 4% formalin after collection, and identified under the microscope after precipitation concentration [35].

Fish were captured using the same fishing gear and methods; the fishing net included a fixed gill net (mesh size: 4~6 cm; mesh height: 0.6~1.5 cm; mesh length: 100~300 m), gill net (mesh sizes: 2.5, 3, 7, and 8 cm; mesh height: 2 m; mesh length: 100~300), and shrimp cage. Sampling started at dusk (approximately 18:00 h) and lasted 12 h through the night. All individuals were identified to species level based on *Fishes of Sichuan* [36]. Fish samples that could not be identified on the spot were fixed with 5~10% formalin solution, the fishing time and place were marked, and species identification was performed after being brought back to the laboratory. In the biological measurement of all catches, body length was accurately measured to 1 mm and body weight to 0.1 g.

*2.3. Data Analysis*

2.3.1. Statistical Analyses

Using Statistical Package for the Social Sciences (IBM SPSS Statistics version 26.0), linear regressions were conducted to investigate relationships among different variables. Student's *t*-test was applied to determine whether changes in water quality indicators in 2018 and 2020 were significant. Correlation analyses with a significance level of $p < 0.05$ were reported as significant [37].

2.3.2. Ecological Types

The fish were then divided according to their lifestyle habits into river–lake migration, resident, and river–sea migration. River–lake migration fish are those that spawn in the flowing waters of rivers and are mainly fattened in lakes. Resident fish live in the same area of river their entire lives to breed and feed, with no obvious migratory behavior. River–sea fish are those that spawn in the flowing waters of rivers and are mainly fattened in the ocean. They can be divided into three types based on feeding habits: herbivorous fish, which feed on higher aquatic plants and lower algae; carnivorous fish, which feed on other fish, benthos, and zooplankton; omnivorous fish, which feed on aquatic plants, aquatic insects, small fish, shrimp, and mollusks [38].

2.3.3. Community Diversity

The Shannon–Wiener index ($H'$) [39], Margalef species richness index ($D$) [39], Pielou's evenness index ($J'$) [39], and Simpson dominance index ($\lambda'$) [39] were used to analyze fish community diversity.

2.3.4. Community Stability

Abundance–biomass comparison curves (ABC curves) were used to measure the stability of community structure and the intensity of interference by environmental factors [40]. In a stable or undisturbed state, the biomass dominance curve always lies above the abundance dominance curve. The two curves intersect as the degree of interference increases and the population becomes moderately disturbed or unstable. When the biomass dominance curve remains below the abundance dominance curve, the community is considered severely disturbed or unstable [41].

Statistical analysis of fish, phytoplankton, and zooplankton data was performed in Excel 2010, OriginPro 8 software (OriginLad, Upton, MA, USA) was used for drawing, and calculations of the diversity index and community structure analysis were carried out using Primer 5.0 software (Plymouth Marine Laboratory, Plymouth, UK).

**3. Results**

*3.1. Spatiotemporal Variations of Water Quality*

Water quality monitoring shows the main elements of plant nutrient pollutants, measured as TN and TP, in the SS and JY reaches. In 2018, the TN content of the SS river reach was 2.2 mg/L, which exceeds the Environmental Quality Standard for Surface Wa-

ter Class V water standard (2.0 mg/L), and it decreased to 1.1 mg/L in 2020, which is between the Environmental Quality Standard for Surface Water Class III water standard (1.0 mg/L) and Class IV water standard (1.5 mg/L). The TP contents in 2018 and 2020 were 0.18 and 0.12 mg/L, respectively, which is lower than the Environmental Quality Standard for Surface Water Class III water standard (0.2 mg/L). The content of organic pollutants ($NO_3-N$, $PO_4^{3-}-P$, and $COD_{Mn}$) and plant nutrient element pollutants (TN, TP, and $NH_4^+-N$) in the SS river reach significantly decreased in 2020 ($p < 0.05$) (Figure 2a). In 2018, the content of TN in the JY river reach was 1.6 mg/L, which is between the Environmental Quality Standard for Surface Water Class IV water standard (1.5 mg/L) and the Class V water standard (2.0 mg/L) and lower than the Environmental Quality Standard for Surface Water Class III water standard (1.0 mg/L). The TP contents in 2018 and 2020 were 0.16 and 0.14 mg/L, respectively, which is lower than the Environmental Quality Standard for Surface Water Class III water standard (0.2 mg/L). The contents of organic pollutants and plant nutrient element pollutants in the JY river reach decreased in 2020. Among them, $PO_4^{3-}-P$ and TP were significantly lower in 2020 ($p < 0.05$) (Figure 2b).

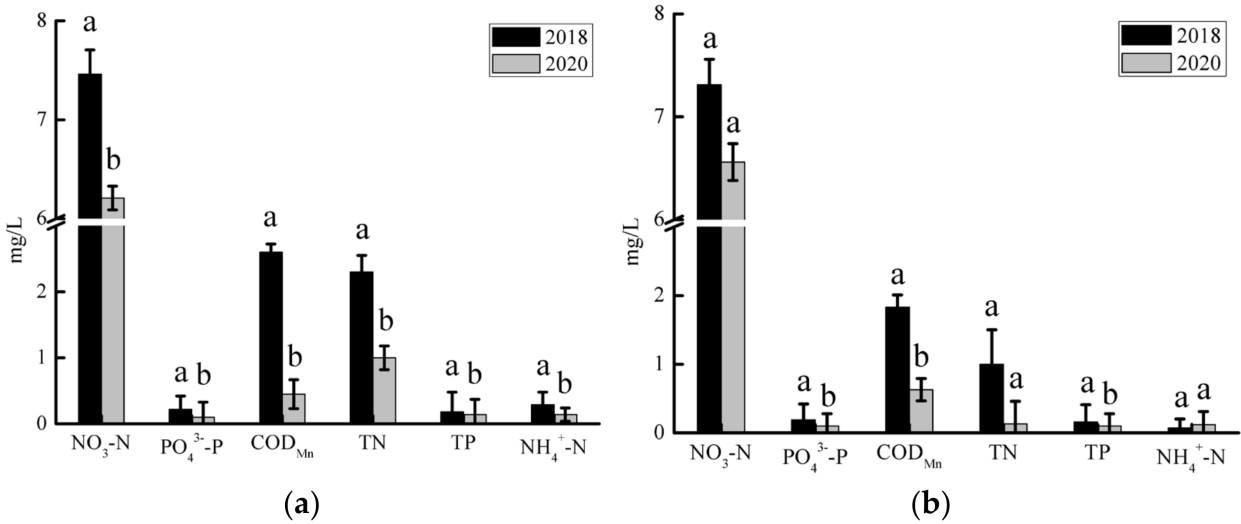

**Figure 2.** Mean values of different water quality parameters in Shishou (**a**) and Jiayu (**b**) river reaches in different years; the associated bars represent standard deviations. The letters above the error bar represent statistically significant differences in the variables ($p < 0.05$).

### 3.2. Spatiotemporal Variations of Phytoplankton and Zooplankton Density

The quantitative results show that the phytoplankton density in the SS river reach was $0.51 \times 10^6$ cells/L in 2018, and the community structure assessment showed mainly species of Bacillariophyta. The phytoplankton density was $0.2 \times 10^6$ cells/L in 2020, a decrease compared with 2018 (Figure 3a), and the community mainly comprised Cyanophyta, Chlorophyta, and Bacillariophyta. The density of zooplankton was 1200 ind/L in 2018, and the community structure was mainly based on Protozoa. The zooplankton density in 2020 was 344 ind/L; thus decreased in 2020 compared with 2018, and the community was mainly dominated by Cladocera and Copepod (Figure 3b). The phytoplankton density in the JY river reach in 2018 and 2020 was $1.6 \times 10^6$ and $0.47 \times 10^6$ cells/L, respectively, showing a decrease with time. The JY rivers reach phytoplankton community structure was mainly based on Cyanophyta, Chlorophyta, and Bacillariophyta (Figure 3c). Here in 2018, the zooplankton density was 1988 ind/L and the community structure was mainly based on Protozoa. The zooplankton density in 2020 was 844 ind/L, showing a decrease with time, and the community then became mainly dominated by Protozoa and Rotifera (Figure 3d).

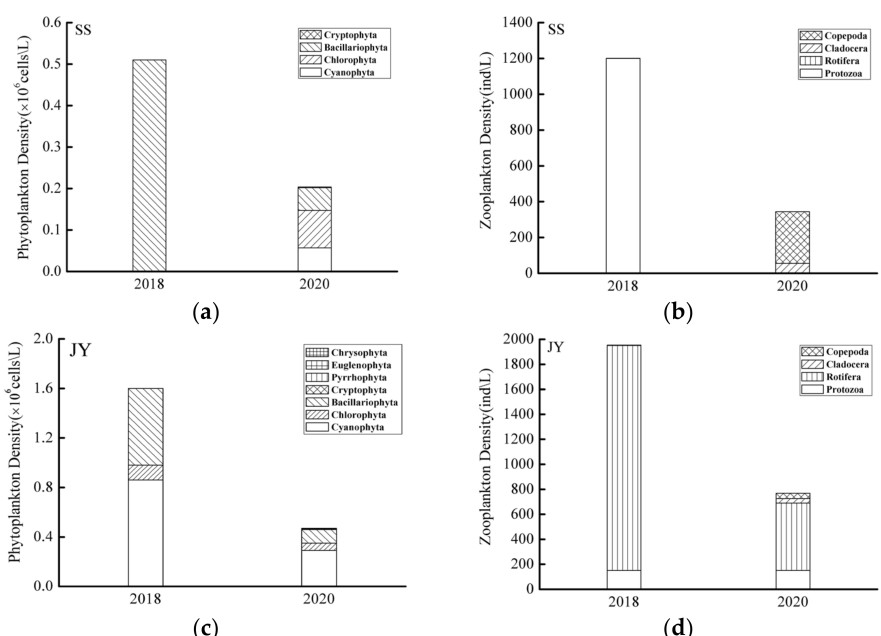

**Figure 3.** Variation characteristics of phytoplankton and zooplankton density in Shishou (**a**,**b**) and Jiayu (**c**,**d**) river reaches, respectively, in different years.

*3.3. Spatiotemporal Variations of the Fish Community Structure and Diversity*

3.3.1. Ecological Type

Based on feeding habits, as indicated in the Supplementary Materials Tables S1 and S2. SS river reach herbivorous fish and carnivorous fish accounted for 34.09% and 23.14%, respectively, of the total in 2020 (Table S2), with an increasing compared with 2018 (Table S1). Omnivorous fish accounted for 42.77% of the total in 2020 (Table S2), which was lower than that in 2018 (Figure 4a) (Table S1). As indicated in the Supplementary Materials Tables S3 and S4. JY river reaches herbivorous fish and carnivorous fish accounted for 10.09% and 69.16%, respectively, of the total in 2020, and the number of omnivorous fish accounted for 20.76% of the total (Table S4). The number of herbivorous fish and carnivorous fish increased compared with 2018 (Table S3), while the number of omnivorous fish decreased (Figure 4b).

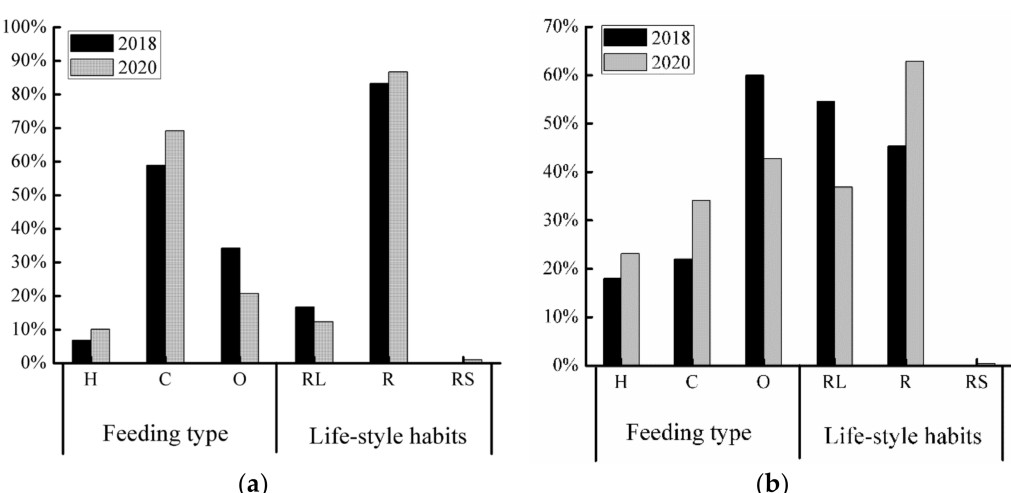

**Figure 4.** Percentage of fish species by ecological types in Shishou (**a**) and Jiayu (**b**) river reaches in different years. #Ecotypes: H, herbivorous; C, carnivorous; O, omnivorous; RL, river–lake migration; R, resident; RS, river–sea migration.

In terms of lifestyle habits, as indicated in the Supplementary Materials Tables S1 and S2. River–lake migration fish in the SS river reach accounted for 36.83% of the total in 2020 (Table S2), with a decrease in the number of rivers–lake migration fish compared with 2018 (Table S1); river–sea migration fish accounted for 0.4% of the total (Figure 4a) (Table S2). The number of migratory fish species increased from 5 to 8 (Table S2). As indicated in the Supplementary Materials Tables S3 and S4. JY reach river–lake migration fish accounted for 12.29% of the total in 2020 (Table S4), with a decrease in the number of rivers–lake migration fish compared with 2018 (Table S3); river–sea migration fish accounted for 1% of the total (Figure 4b) The number of migratory fish species increased from 6 to 9 (Table S4).

### 3.3.2. Community Stability

Statistical analysis revealed that the biomass dominance curve of fish communities in the SS river reach in 2018 was above the quantitative dominance curve, with a $W-$statistic of 0.058 (Figure 5a). These curves intersected in 2020, with a W-statistic of 0.026 (Figure 5b). The biomass dominance curves and population dominance curves of fish communities in the JY river reach in 2018 and 2020 all intersected to different degrees, with a W-statistic of 0.016 and $-0.018$, respectively (Figure 5c,d).

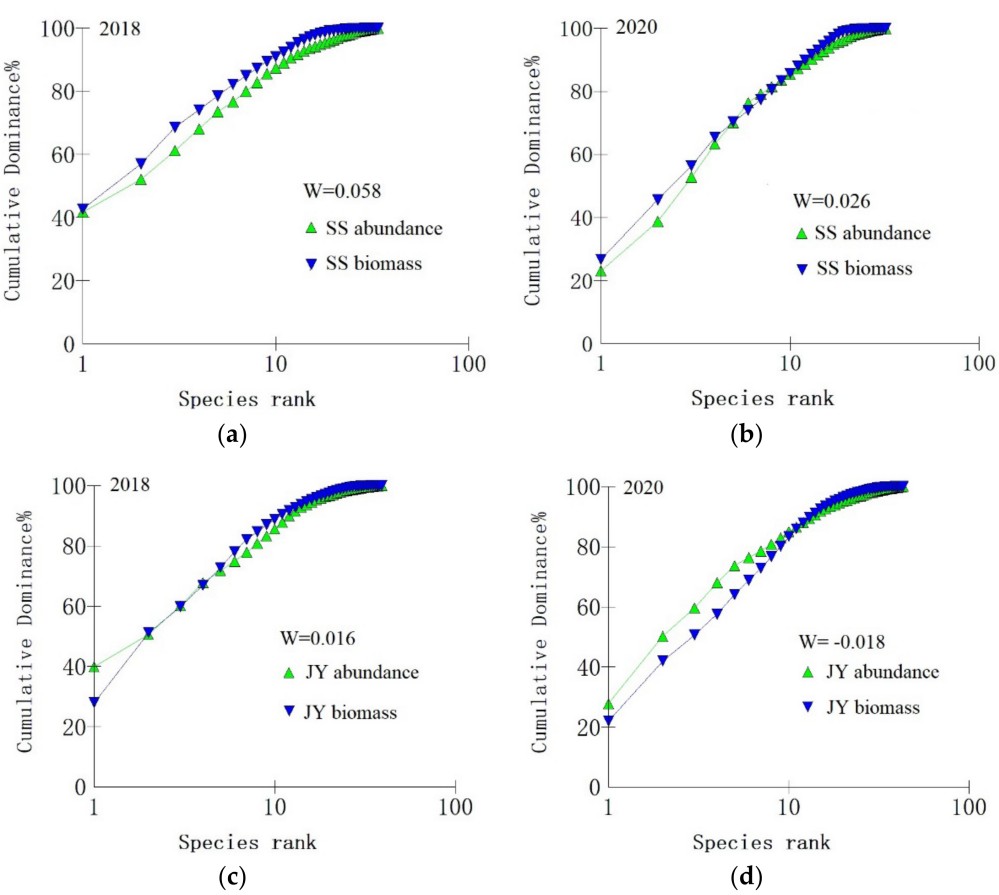

**Figure 5.** Abundance−biomass comparison curves (ABC) and W−statistics of fish communities in the Shishou (**a**,**b**) and Jiayu (**c**,**d**) river reaches in different years.

### 3.3.3. Diversity Characteristics

Species diversity indexes were as follows: In the SS river reach, $D = 5.09$, $H' = 3.64$, $J' = 0.73$, and $\lambda' = 0.12$ in 2020. Comparison with the values in 2018 shows that $D$, $H'$, and $J'$ have increased while $\lambda'$ has decreased (Figure 6a). In the JY river reach, $D = 6.05$, $H' = 3.89$, $J' = 0.67$, and $\lambda' = 0.15$ in 2020. Comparison with the values in 2018 shows that $D$, $H'$, and $J'$ have increased while $\lambda'$ has decreased (Figure 6b). The $D$ and $H$ of the SS and JY river

reaches were higher in 2020 than in 2018, indicating that species richness and diversity were high in 2020.

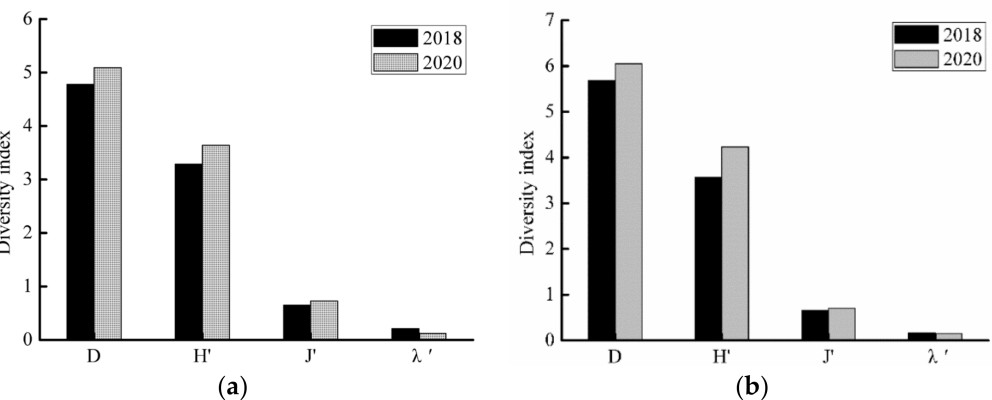

**Figure 6.** Variation in values of the fish community diversity index in Shishou (**a**) and Jiayu (**b**) river reaches in different years. *D*: Margalef species richness index; *H′*: Shannon−Wiener diversity index; *J′*: Pielou's evenness index; *λ′*: Simpson dominance index.

## 4. Discussion

### 4.1. Possible Reasons for Water Quality Improvements

Human activities impact the surrounding ecological environment, altering the nitrogen cycle in aquatic systems, changing hydrological circulation, deteriorating river water quality, and determining the carrying capacity of the water environment to a certain extent [42]. China's rapid economic development has placed considerable pressure on aquatic environments, resulting in the widespread deterioration of water quality [43]. COVID-19 lockdowns dampened the high intensity of human activity, which had beneficial implications for water quality.

The results of this study show that levels of both organic contaminants and plant nutrient contaminants in the river decreased during COVID-19 lockdown periods in 2020. Among them, $PO_4^{3-}-P$, $COD_{Mn}$, and TP were significantly decreased after lockdown ($p < 0.05$). Studies have shown that discharge from industries and domestic effluents are the major contributors to the increased pollution load in river systems [44]. Industrial pollution is an important source of $NO_3-N$, $PO_4^{3}-P$, and $COD_{Mn}$ in Chinese rivers [45]. TN and TP are the main causes of water eutrophication, and their presence in sewage is mainly derived from domestic effluent, garbage, feces, etc. [46]. The COVID-19 lockdowns resulted in the shutdown of many industries, and people were asked to stay at home, resulting in reduced discharge of industrial and domestic effluents and improved river water quality. Some studies have reported the positive impacts of COVID-19 lockdowns on water quality. For example, reductions in industrial effluent discharges and vessel traffic led to improved water transparency in the Venetian Lagoon, Italy [47]. Recent research reported that less domestic wastewater discharge due to COVID-19 lockdowns led to lower nitrate ($NO_3-N$) concentrations, and the Ganges water has been revived to the extent that it is now potable [48]. Studies have shown that improvements in air quality might also have beneficial impacts resulting in improved water quality. COVID-19 lockdowns reduced atmospheric pollution, and atmospheric deposition is an important source of water nutrients, which can further increase COD by enhancing primary production [49].

The results of our study show a reduction in industrial and domestic effluent discharge, substantial decreases in the production of air pollutants, decreased atmospheric nutrient deposition, and a decrease in organic contaminants and plant nutrient contaminants in the examined river areas due to COVID-19 lockdowns; among them, $PO_4^{3-}-P$, $COD_{Mn}$, and TP have significantly decreased, improving the quality of river water.

*4.2. Plankton Response to Water Quality Changes*

Plankton act as primary producers in aquatic ecosystems, providing bait and the necessary energy for other organisms [50]. In an aquatic ecosystem, the growth of plankton is closely related to water environment factors. The seasonal succession of the plankton community structure can represent the state of the water environment, and the spatiotemporal change in the water environment can cause changes in plankton density, biomass, and diversity [51]. Among the various environmental water factors, WT, TN, TP, and pH are the main abiotic factors affecting the plankton community structure [52].

The results of this study show that the phytoplankton and zooplankton density of the SS and JY river reaches in 2020 were decreased compared with in 2018. We speculate that this may be related to reductions in organic pollutants and plant nutrient pollutants in the water body. Studies have shown that $NO_3-N$, $PO_4^{3-}-P$, and WT are the most obvious environmental factors affecting phytoplankton growth in spring and summer [53]. $COD_{Mn}$ is an important indicator for monitoring the organic matter content in the water environment; organic matter in water can promote the growth and reproduction of algae, which is positively correlated with phytoplankton growth [54]. TN and TP are the material basis of algae growth [55]. One study found that an increase in TN and TP contents in the lower Rhine River promoted an increase in phytoplankton species and density [56]. $NH_4^+-N$ is the form of elemental nitrogen that can be directly absorbed phytoplankton [57]. During COVID-19 lockdowns, reductions in industrial and domestic effluent discharge led to a reduction in the nutrient supply, which inhibited algae growth and caused a decrease in the phytoplankton density. As primary consumers in the food chain, zooplankton mainly feed on phytoplankton, bacteria, and detritus, and the abundance of zooplankton can be influenced by phytoplankton via a bottom-up effect.

*4.3. Possible Reasons for Changes in Fish Diversity and Community Structure*

Biodiversity, especially its decline due to human activities, is a global concern [58]. Water pollution is one of the main threats to the survival and reproduction of all fish [59]. Overfishing is another of the main threats to fish, which has affect their community structure, reduces the number of supplementary populations, and has led to more species becoming increasingly threatened with extinction [60]. Sand dredging ships, which increases water turbidity and leave silt suspended in the water, may block fish gills or breathing holes and thus affect the normal breathing of fish, which is particularly harmful to hatching fry [61].

The results of this study show that in the SS and JY river reaches in 2020, $H' = 3.64$ and 3.89, respectively, which is an increase compared with 2018. Changes in environmental factors are an important factor causing changes in fish diversity. One study found that high concentrations of $NH_4^+-N$, TN, and TP in water lead to a decrease in fish diversity, and water pollution adversely affects the species composition and diversity of fish in rivers [62]. During COVID-19 lockdowns, industrial and domestic effluent discharge was reduced, river water quality improved, and fish diversity increased. Overfishing is an important threat to fish diversity, and studies have shown that community diversity is closely related to disturbance intensity, where diversity is highest when the disturbance is at a moderate intensity and lowest when the disturbance is at two different levels [63]. COVID-19 lockdowns required people to stay at home, limiting anthropic activities and therefore reducing stress due to fishing. The external disturbance to fish resources was weakened and fishing intensity was effectively reduced, resulting in higher fish diversity. On the other hand, fishing intensity decreased, which had a protective effect on a large number of reproductive parents, thus increasing the output of fry and therefore species diversity and richness [64].

The results show that carnivorous fish in the SS and JY river reaches accounted for 23.14% and 69.16% of the total in 2020, respectively, which were increased in numbers compared to 2018. The fish community structure is closely related to the environment of the aquatic ecosystem in which it lives [65]. One study found that under conditions of declining environmental water quality, omnivorous fish have a greater survival advantage due to

more diverse food sources, a wider range of ecological niches, and stronger adaptability to the environment [66]. During COVID-19 lockdowns, the reduced nutrient supply, as measured by TN and TP, led to lower primary productivity in rivers, a reduced supply of fish feed, intensified competition among fish communities, and increased populations with the dominance of high trophic level carnivorous fish. In addition, river–sea migratory fish accounted for 3.4% and 4.2% of the total in SS and JY river reaches in 2020, respectively, increasing compared to 2018. This suggests that COVID-19 lockdowns reduced fishing intensity and thus the risk of river–sea migration fish being caught, effectively allowing them to complete long-distance spawning migration, which results in increased populations. The results of the ABC curve—both the abundance dominance curve and the biomass dominance curve—show that the SS and JY river reaches had different degrees of intersection in 2020, indicating that the fish community was mainly dominated by small-sized species with a short life cycle. This suggests that COVID-19 lockdowns, through limitation of anthropic activities and reduced fishing efforts and harvest, resulted in increases in the number of supplementary populations. In a similar vein, one study found that the summer close season can effectively increase the amount of juvenile fish supplementary population resources [67].

The results of this study suggest that COVID-19 lockdowns have reduced the interference of human activities on fishery resources, wherein the fish community became mainly dominated by small-sized species with a short life cycle, and the number of supplementary populations increased. The reduction in fishing pressure has contributed to an increase in the number of rivers–sea migratory fish. At the same time, improvements in river water quality have contributed to an increase in the number of carnivorous fish. The fish community structure showed a tendency toward higher complexity and fish diversity.

## 5. Conclusions

The results of this study indicate that COVID-19 lockdowns had positive impacts on the Yangtze River ecosystem. During the entire course of lockdowns, industrial and domestic effluent discharges decreased, and levels of organic contaminants and plant nutrient contaminants in the river also decreased; among them, $PO_4^{3-}-P$, $COD_{Mn}$, and TP were significantly decreased ($p < 0.05$), and river water quality improved. The decrease in nutrient supply as indicated by TN and TP inhibited the algae growth and caused the density of phytoplankton to decrease. Zooplankton mainly feed on phytoplankton, bacteria, and detritus, and phytoplankton can thus affect the abundance of zooplankton through a bottom-up effect, wherein decreases in phytoplankton density lead to decreased zooplankton density. The decrease in plankton density has led to lower primary productivity in rivers, reduced fish feed supplies, intensified competition among fish populations, and increased populations dominated by high trophic level carnivorous fish. In addition, fishing intensity was effectively reduced by COVID-19 lockdowns, which limited anthropic activities, and the fish community became mainly dominated by small-sized species with a short life cycle, with the number of supplementary fish populations also increasing. The reduction in fishing pressure has contributed to an increase in the number of rivers–sea migratory fish. The fish community structure showed a tendency toward higher complexity and fish diversity. Ecological protection issues in the Yangtze River have recently received considerable attention. COVID-19 lockdowns have provided respite for the retrogressive Yangtze River ecosystem.

**Supplementary Materials:** The following supporting information can be downloaded at: https: //www.mdpi.com/article/10.3390/w14223622/s1, Table S1: Species composition and ecological types of fish in Shishou river reach in 2018; Table S2: Species composition and ecological types of fish in Shishou river reach in 2020; Table S3: Species composition and ecological types of fish in Jiayu river reach in 2018; Table S4: Species composition and ecological types of fish in Jiayu river reach in 2020.

**Author Contributions:** Conceptualization, H.Y. and H.Z.; Data curation, D.F. and H.Y.; Formal analysis, D.F. and H.Y.; Funding acquisition, H.D. and H.Y.; Investigation, J.Y. and D.F.; Methodology,

D.F. and H.Y.; Project administration, Q.W., H.Z. and H.Y.; Software, D.F.; Visualization, D.F.; Writing-original draft, D.F.; Writing-review & editing, D.F., H.Y. and H.Z. All authors have read and agreed to the published version of the manuscript.

**Funding:** This work was supported by the Central Public-Interest Scientific Institution Basal Research Fund, Chinese Academy of Fishery Sciences (Grant numbers YFI202201, 2020TD08); Project of Yangtze Fisheries Resources and Environment Investigation from the Ministry of Agriculture and Rural Affairs, China (Grant Numbers CJDC-2017-14).

**Institutional Review Board Statement:** Not applicable.

**Informed Consent Statement:** Not applicable.

**Data Availability Statement:** Not applicable.

**Conflicts of Interest:** The authors declare no conflict of interest.

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
