# Peer review of "COVID-19 Lockdowns Promoted Recovery of the Yangtze River’s Aquatic Ecosystem"

_water, doi:10.3390/w14223622_

Round 1

Reviewer 1 Report

General comment

The submitted MS represents an interesting study on the quality of freshwater habitat regarding anthropogenic pressure. I find it suitable for publishing in Water, but after some minor changes are done. 

Specific comments

Pg. 2 - Authors name and year - Tenualosa ilisha

Pg. 3 – provide some literature sources of mentioned surveys: The Yangtze River Fisheries Research Institute, Chinese Academy of Fishery Sciences, has been conducting year-round surveys at these locations since 2017 (ref!).

Pg. 3 – cite studies that confirm: They are also one of the most heavily polluted urban-based river systems (ref.!).

Figure 1. add the name of the study area in the caption of Figure 1. as well as explanations of abbreviations (SS and JY). Keep in mind that each figure should be self-explanatory, and separate from the main text.

In situ in italics - in situ

It is not clear – was all sampling conducted at the same time? Pyhto-and zooplankton and fishes? And the chemical properties of water? Please clarify.

Pg. 4. Nowadays there is no need to have formulas of widely used indices as Shannon, Pielou and Margalef presented in the MS. Please delete them.

Also, delete the formula of abundance-biomass comparison curves and shorten the text; thus add in the brackets (see more  in 33).

Figure 2 – also, you cannot have abbreviations of study sites, it is not clear where the study was performed. It must be self-explanatory. Please give full sites' names as well as an explanation of other abbreviations such as TN, TP...; the sentence The letters above the error bar represent statistically significant differences in the variables is not clear. Clarify, please.

Figures 3, 4, 5, 6 – also full study sites' names

As diversity indices were used it is obligatory to provide a species list with habitats and years of sampling. It can be in the table as a supplement material to the MS. Thus, the beginning of 3.3. the section should start with results about the total number of recorded species, and then per habitat and per year.  Also, point out the dominant and the most abundant fish species.

Discussion

Pg. 10 – there is no need to again repeat the result values of phytoplankton density in the second paragraph of section 4.2.; just comment on it, discuss it in relation to other studies or etc.

Author Response

Dear Reviewer

On behalf of my co-authors, we thank you very much for allowing us to revise our manuscript, we appreciate the editor and reviewers very much for their positive and constructive comments and suggestions on our manuscript entitled “COVID-19 Lockdowns Promoted Recovery of the Yangtze River’s Aquatic Ecosystem”. (ID: water-2006508).

We have studied the reviewers’ comments carefully and have made revisions which are marked in red in the paper. We have tried our best to revise our manuscript according to the comments. I am attaching the revised version, which we would like to submit for your kind consideration.

We want to express our great appreciation to you and the reviewers for your comments on our paper. We are looking forward to hearing from you.

Best wishes,

Yours sincerely,

Dongdong Fang

Email: fangdong19910111@126.com

Reviewer 2 Report

The study reports interesting results on the monotoring activity carried out for the selected river before and during the pandemic. The paper is well structured and the research proved that COVID-19 lockdowns had positive impacts on the Yangtze River ecosystem.

I have only one comment, the introduction section is too short. Pheraps the literature review might be improved. Furthermore, since the the focus of the paper is SARS-CoV-2, before describing the effects of the lockdowns, some comments on the wastewater-based epidemiology (WBE) approach and on fate of Sars-CoV- in wastewater might be mentioned. Following an example of study that might be included within the description:

1  - SARS-CoV-2 from Urban to Rural Water Environment: Occurrence, Persistence, Fate and Influence on Agriculture Irrigation. A Review. G. Mancuso, G. D. Perulli, S. Lavrnić, B. Morandi, A. Toscano | Water 13(6) (2021) 764.

Author Response

Dear Reviewer

On behalf of my co-authors, we thank you very much for giving us an opportunity to revise our manuscript, we appreciate editor and reviewers very much for their positive and constructive comments and suggestions on our manuscript entitled “COVID-19 Lockdowns Promoted Recovery of the Yangtze River’s Aquatic Ecosystem”. (ID: water-2006508).

We have studied reviewers’ comments carefully and have made revision which marked in red in the paper. We have tried our best to revise our manuscript according to the comments. Attached please find the revised version, which we would like to submit for your kind consideration.

We would like to express our great appreciation to you and reviewers for comments on our paper. Looking forward to hearing from you.

Best wishes,

Yours sincerely,

Dongdong Fang

Email: fangdong19910111@126.com

Reviewer 3 Report

Actually, I have no serious concerns about this paper, so it might be recommended for publication in current version

Author Response

Dear Reviewer

On behalf of my co-authors, we thank you very much for allowing us to revise our manuscript, we appreciate the editor and reviewers very much for their positive and constructive comments and suggestions on our manuscript entitled “COVID-19 Lockdowns Promoted Recovery of the Yangtze River’s Aquatic Ecosystem”. (ID: water-2006508).

We have studied the reviewers’ comments carefully and have made revisions which marked in red in the paper. We have tried our best to revise our manuscript according to the comments. I am attaching the revised version, which we would like to submit for your kind consideration.

We want to express our great appreciation to you and the reviewers for your comments on our paper. We are looking forward to hearing from you.

Best wishes,

Yours sincerely,

Dongdong Fang

Email: fangdong19910111@126.com

Reviewer 4 Report

   The article concerns relevant issues. The logic of the research is clear , methods used are appropriate, conclusions are reasonable.

Author Response

(The authors gave the same response as above.)
